# Psychological predictors of vaccination intentions among U.S. undergraduates and online panel workers during the 2020 COVID-19 pandemic

Suryaa Gupta[¤a‡], Shoko Watanabe[‡], Sean M. Laurent[¤b]*

Department of Psychology, University of Illinois at Urbana-Champaign, Champaign, Illinois, United States of America

¤a Current address: National Institute for Child Health and Human Development, Bethesda, Maryland, United States of America
¤b Current address: Department of Psychology, Pennsylvania State University, University Park, Pennsylvania, United States of America
‡ SG and SW share co-first authorship and contributed equally to this work.
* seanmlaurent@gmail.com

**Data Availability Statement:** The study design, sample size, and analyses were preregistered here: https://osf.io/rycjd. All data and analysis codes are available here: https://osf.io/rd2s6/. Study materials

## Abstract

### Objective

Availability of safe and effective vaccines against COVID-19 is critical for controlling the pandemic, but herd immunity can only be achieved with high vaccination coverage. The present research examined psychological factors associated with intentions to receive COVID-19 vaccination and whether reluctance towards novel pandemic vaccines are similar to vaccine hesitancy captured by a hypothetical measure used in previous research.

### Method

Study 1 was administered to undergraduate students when COVID-19 was spreading exponentially (February-April 2020). Study 2 was conducted with online panel workers toward the end of the first U.S. wave (July 2020) as a pre-registered replication and extension of Study 1. In both studies, participants (total $N = 1,022$) rated their willingness to receive the COVID-19 vaccination and to vaccinate a hypothetical child for a fictitious disease, and then responded to various psychological measures.

### Results

In both studies, vaccination intentions were positively associated with past flu vaccine uptake, self-reported vaccine knowledge, vaccine confidence, and sense of collective responsibility. Complacency (not perceiving disease as high-risk), anti-vaccine conspiracy beliefs, perceived vaccine danger, and mistrust in science/scientists were negative correlates of vaccination intentions. Constraints (psychological barriers), calculation (extensive information-searching), analytical thinking, perceived disease vulnerability, self-other overlap, and conservatism were weakly associated with vaccination intentions but not

and supplementary analyses are reported in the Supporting Information file (Online Supplemental Materials).

**Funding:** This research was supported, in part, by a grant awarded to S. G. from the Office of Undergraduate Research, University of Illinois at Urbana-Champaign. The funder had no role in study design, data collection and analysis, decision to publish, or preparation of the manuscript.

**Competing interests:** The authors have declared that no competing interests exist.

consistently across both studies or vaccine types. Additionally, similar factors were associated with both real and hypothetical vaccination intentions, suggesting that conclusions from pre-COVID vaccine hesitancy research mostly generalize to the current pandemic situation.

## Conclusion

Encouraging flu vaccine uptake, enhancing confidence in a novel vaccine, and fostering a sense of collective responsibility are particularly important as they uniquely predict COVID-19 vaccination intentions. By including both actual pandemic-related hesitancy measures and hypothetical hesitancy measures from past research in the same study, this work provides key context for the generalizability of earlier non-pandemic research.

## Introduction

The rapid spread of COVID-19 has resulted in a global health crisis. As of November 2021, over 767,000 American lives have been lost [1]. For most of 2020, despite safety precautions (e.g., physical distancing, frequent hand-washing), the lack of effective antiviral treatment or vaccines resulted in a serious burden to the health care system. Additionally, resulting from local and state governments issuing lockdowns, the U.S. economy has been strongly negatively impacted [2]. Effective COVID-19 vaccines were hoped to both reduce detrimental health impacts and facilitate economic recovery by lifting pandemic restrictions. However, vaccines can only protect communities when enough people choose to vaccinate [3]. The early stages of the pandemic provided an unexpected opportunity to study vaccination intentions when effective treatments or vaccines against COVID-19 had not yet been developed or deployed, and social norms regarding the new vaccine had not yet been established. Whereas vaccine hesitancy research prior to the pandemic examined attitudes about vaccination for diseases that are hypothetical [4, 5], relatively low-threat (e.g., seasonal flu) [6], or mostly eradicated in the U.S. (e.g., polio) [7], the present research examined which psychological factors are associated with intention to receive a novel, pandemic vaccine in a context where no one was immune to a highly contagious and life-threatening disease.

Building on prior research that has identified individual, contextual, and vaccine-specific issues [7–10], the goal of our research was to examine whether these predictors generalize to a novel situation in which individuals decide whether to receive a newly vaccine designed to fight a previously unknown disease. Because the COVID-19 vaccine was developed and authorized in record time amidst unprecedented socioeconomic disruptions, there may be unique factors influencing vaccination intentions. Additionally, psychological research prior to the pandemic has documented anti-vaccine conspiracy beliefs and perceived dangers of vaccines as negative correlates of vaccination intentions, but findings were based on attitudes toward vaccinating a hypothetical child for a fictitious disease [4, 5, 11]. Using hypothetical scenarios is effective for conducting experiments because researchers can tightly control specific information about diseases or vaccines, but such methods suffer from ecological validity concerns. Thus, by measuring vaccination intentions for both COVID-19 (a real disease) and a fictitious child disease, we sought to examine whether previously identified factors associated with hypothetical immunization decisions are also predictive of real vaccination intentions.

While this article was under review, various studies on COVID-19 vaccination intention were published (see [12] for a review). Below, we briefly present findings from selected U.S. studies assessing similar measures as the current study. A cross-sectional study conducted in October 2020 using the 5C psychological antecedents of vaccination [13], a previously validated measure the current study also administered, found that perceived benefits (vaccine confidence and collective responsibility) as well as barriers to vaccination (complacency and constraints) were independently associated with COVID-19 vaccination intention [14]. Other cross-sectional surveys conducted in May-June 2020 showed that willingness to receive COVID-19 vaccine was associated with past flu vaccination behavior, perceived likelihood of infection, perceived severity of COVID-19, altruism, liberal political leaning, and having higher income [15–17]. Additionally, a longitudinal study conducted in March and July of 2020 showed increasing hesitancy over time and that conspiracy beliefs were inversely related to perceptions of vaccine safety and COVID-19 vaccination intention [18]. Another longitudinal survey conducted in March-August 2020 showed diverging ideological trajectories of COVID-19 vaccination intentions, where intentions declined among Republicans but not Democrats [19]. These studies highlight the importance of vaccination messages for different people and situations, and that decisions to accept a novel vaccine are complex and multi-faceted [20]. The current research adds to the collection of studies aimed to improve our understanding of attitudes about vaccinations.

## The present research

The present research examined vaccine hesitancy when the pandemic was spreading exponentially in the U.S and vaccine development was initiating (Study 1; February-April 2020) and during Phase II clinical trials at the end of the first U.S. wave (Study 2; July 2020). Specifically, our research aimed to predict vaccination intentions for COVID-19 and for a hypothetical childhood disease (dysomeria) used in past research conducted before the pandemic (see Measures). We hypothesized that both COVID-19 and dysomeria vaccination intentions would be positively associated with confidence and collective responsibility aspects of the 5C psychological antecedents of vaccination [13], vaccine knowledge, analytical thinking, perceived disease vulnerability, self-other overlap, and prior flu vaccination. We also hypothesized that vaccination intentions would be negatively associated with complacency, constraints, and calculation aspects of the 5C, anti-vaccine conspiracy beliefs, perceived vaccine danger, mistrust in science/scientists, and political conservatism.

Verbatim materials and supplementary analyses are reported in the S1 File. Beyond what is reported here, additional measures were collected for both samples and that are reported only in the S1 File. For succinctness, analyses reported here focus on measures that were collected for both samples, but interested readers are encouraged to consult the S1 File for results of other highly relevant variables (e.g., media consumption). All data and analysis codes are available at https://osf.io/rd2s6/. Data were analyzed using R version 4.0.5 [21]. This research was approved by the Institutional Review Board at the University of Illinois, Urbana-Champaign. Waiver of documentation of informed consent was approved for this study.

## Study 1

### Participants and procedure

Study 1 participants included 346 students recruited from the undergraduate course-credit subject pool of the University of Illinois at Urbana-Champaign in Spring 2020. Undergraduate students in psychology courses who are 18 or older have the option to enroll in the subject pool to participate in research studies in exchange for course credit or to complete an

alternative assignments with a similar duration. Participants in the subject pool voluntarily and individually sign up to complete studies and can withdraw from studies at any time. This data collection method is a standard practice in psychology [22, 23]. Sample size was determined by availability of credits. For Study 1, data collection began on February 24, 2020, and ended on April 21, 2020. After reading the informed consent and consenting to participate, participants responded to a number of measures before providing demographics and being debriefed. The university transitioned to fully remote learning on March 23, but the study was always administered online.

Participants' data were excluded from analyses if they failed one or more attention-check questions, and/or spent $M \leq 10s$ per screen. After exclusions (see the S1 File for details), 308 participants remained (76.95% female, $M_{age}$ = 19.65, $SD_{age}$ = 1.35). Sensitivity analyses showed that with $\alpha$ = .05 (two-tailed), we had 80% power to detect $r$ = |0.16|. Reported racial/ethnic identities were Asian/Asian American (18.51%), Black/African American (5.52%), Hispanic/Latino (17.21%), White/European American (49.35%), more than one (7.14%), and other/prefer not to say (2.27%).

## Measures

Unless noted, all items used 7-point scales ranging from 1 = *Strongly disagree* to 7 = *Strongly agree*.

**Vaccination intention (COVID-19).** Participants read CDC-provided information about COVID-19, including total U.S. cases and deaths and that there were no antiviral treatments or vaccines yet available (see S1 File for details). Participants were asked whether they would receive the COVID-19 vaccine if it were available. Response options were: "*I would...*" 1 = *not receive the vaccination even if it's free ($0)*, 2 = *receive the vaccination only if it's free ($0)*, 3 = *pay up to $10 to receive the vaccination*, 4 = *pay up to $25...*, 5 = *pay up to $50...*, 6 = *pay up to $100...*, and 7 = *pay more than $100 to receive the vaccination*. We chose this measure because willingness to incur greater financial cost for a novel vaccine should reflect higher commitment to being vaccinated.

**Vaccination intention (dysomeria).** To assess vaccination intention for a childhood disease, participants read about *dysomeria*—a (fictitious) disease spread by droplet infection, causing serious symptoms [11, 24]. Participants were asked to imagine that they had an 8-months-old infant. They were informed that vaccination against dysomeria was recommended by the CDC but that adverse events were reported 12% of the time. Participants then indicated their intention to vaccinate their hypothetical child (1 = *Definitely not vaccinate*, 7 = *Definitely vaccinate*).

**Past flu vaccination uptake.** Participants indicated their past flu vaccination uptake by responding to the question: "This past season, did you receive the flu shot?" with response options: "Yes," "No," and "prefer not to answer." Responses were coded such that 0 = No/prefer not to answer and 1 = Yes.

**Vaccine knowledge.** Three questions assessed self-rated vaccine knowledge (e.g., "Compared to your peers, how much do you know about how vaccines work?") using 7-point scales (1 = *I am not at all knowledgeable*, 7 = *I am extremely knowledgeable*).

**5C psychological antecedents of vaccination.** The 5C Psychological Antecedents of Vaccination measure [13] contains five subscales with three items each: confidence (e.g., "I am completely confident that vaccines are safe"), collective responsibility (e.g., "Vaccination is a collective action to prevent the spread of diseases"), complacency (e.g., "Vaccination is unnecessary because vaccine-preventable diseases are not common anymore"), constraint (e.g., "Everyday stress prevents me from getting vaccinated"), and calculation (e.g., "When I think

about getting vaccinated, I weigh benefits and risks to make the best decision"). Items in each subscale were averaged to form composite scores. A programming error led to one item from the 5C's preliminary scale being presented and was omitted from Study 1 analysis. Study 2 used the correct item from the final scale.

**Anti-vaccine conspiracy beliefs.**   Anti-vaccine conspiracy beliefs [11] were assessed with seven items (e.g., "Immunizations allow governments to track and control people"). One item from the original measure regarding the flu vaccine was not included due to programming error.

**Perceived vaccine danger.**   Perceived dangers of vaccines [11] were assessed with eight items (e.g., "I feel uncertain about the potential side-effects of immunizations").

**Mistrust in science/scientists.**   To assess mistrust in science/scientist, we drew five items with the highest inter-item correlations from the Trust in Science and Scientists Inventory [25]. These items (e.g., "Scientists ignore evidence that contradicts their work.") were measured on 5-point scales (1 = *Strongly disagree*, 5 = *Strongly agree*).

**Analytical thinking.**   The ten highest-loading items (e.g., "I enjoy intellectual challenges") of the rational subscale of the Rational-Experimental Multimodal Inventory (REIm) [26] were used to assess analytic thinking (1 = *Completely false*, 5 = *Completely true*).

**Perceived disease vulnerability.**   Perceived disease vulnerability was measured with the 7-item infectibility subscale (e.g., "I have a history of susceptibility to infectious disease") of the Perceived Vulnerability to Disease Scale [27].

**Self-other overlap.**   An adapted version of the Inclusion-of-Other-in-the-Self scale [28] asked participants to select one of seven Venn-diagrams that best represented their relationship with acquaintances. Higher numbers indicate greater self-other overlap.

**Ideological conservatism.**   Participants indicated their ideological conservatism with one item: "Where would you place yourself on the following ideological spectrum?" (1 = *Extremely liberal*, 2 = *Moderately liberal*, 3 = *Slightly liberal*, 4 = *Middle of the road*, 5 = *Slightly conservative*, 6 = *Moderately conservative*, 7 = *Extremely conservative*).

## Results

**Descriptive statistics.**   A non-negligible proportion of participants from Study 1 indicated hesitancy regarding the COVID-19 vaccine. Specifically, 5.52% indicated they would not receive the vaccine even if it were free, and 6.49% indicated they would receive the vaccine only if it cost $0. The amount of money participants were willing to pay for a COVID-19 vaccine was: up to $10 (6.82%), up to $25 (14.94%), up to $50 (18.18%), up to $100 (15.58%), and more than $100 (32.47%). For the hypothetical dysomeria vaccine, hesitancy was less obvious with only 1.30% indicating that they would "definitely not vaccinate" the child and the majority (62.01%) endorsing the highest end of the scale indicating that they would "definitely vaccinate" the child. For prior vaccination behavior, 56.82% reported receiving the seasonal flu shot.

**Bivariate correlations.**   Table 1 shows means, standard deviations, and correlations between the vaccination intention variables (COVID-19 and dysomeria) and predictor variables. The full correlation matrices of all measured variables are available in the S1 File. COVID-19 and dysomeria vaccination intentions were positively correlated ($r = .31$, $p < .001$). The correlational analyses revealed similar results for both types of vaccination (see Table 1) and support most of our hypotheses. Specifically, past flu vaccine uptake, vaccine knowledge, vaccine confidence, and collective responsibility consistently emerged as positive correlates of COVID-19 and dysomeria vaccination intentions. Complacency, constraint, calculation, anti-vaccine conspiracy beliefs, perceived vaccine danger, and mistrust in science/scientists were all negatively associated with.

**Table 1. Reliability, means, and standard deviations of variables, and correlations between vaccination intention outcome variables (COVID-19 and dysomeria) and predictor variables for Study 1 and Study 2.**

| Variables | Study 1 (undergraduates) | | | | | Study 2 (online panel workers) | | | | |
|---|---|---|---|---|---|---|---|---|---|---|
| | $\alpha/r$ | M | SD | COVID | Dysomeria | $\alpha/r$ | M | SD | COVID | Dysomeria |
| COVID-19 vaccine intention | - | 5.10 | 1.83 | - | .31** | - | 4.57 | 1.96 | - | .45** |
| Dysomeria vaccine intention | - | 6.36 | 1.09 | .31** | - | - | 5.65 | 1.82 | .45** | - |
| Past flu vaccine uptake | - | 0.57 | 0.50 | .25** | .18* | - | 0.56 | 0.50 | .29** | .27** |
| Vaccine knowledge | .88 | 4.97 | 1.17 | .22** | .19** | .92 | 4.93 | 1.37 | .21** | .23** |
| 5C-Confidence | .61 | 5.74 | 0.95 | .41** | .42** | .73 | 5.05 | 1.45 | .50** | .56** |
| 5C-Collective responsibility | .44 | 6.42 | 0.93 | .38** | .49** | .71 | 5.73 | 1.34 | .36** | .51** |
| 5C-Complacency | .52 | 2.04 | 1.05 | -.23** | -.37** | .82 | 2.39 | 1.51 | -.22** | -.42** |
| 5C-Constraint | .69 | 1.94 | 1.08 | -.19** | -.27** | .83 | 2.33 | 1.49 | -.06 | -.22** |
| 5C-Calculation | .78 | 3.86 | 1.66 | -.20** | -.29** | .77 | 5.30 | 1.38 | -.07 | -.03 |
| Conspiracy beliefs | .83 | 1.95 | 0.90 | -.32** | -.34** | .91 | 2.57 | 1.43 | -.27** | -.45** |
| Perceived vaccine danger | .90 | 2.78 | 1.18 | -.33** | -.46** | .92 | 3.39 | 1.44 | -.35** | -.45** |
| Mistrust in science/scientists | .86 | 1.77 | 0.64 | -.27** | -.24** | .94 | 2.22 | 1.05 | -.33** | -.42** |
| Analytical thinking | .83 | 3.78 | 0.58 | .08 | .17* | .88 | 3.67 | 0.77 | .11* | .10 |
| Disease vulnerability | .88 | 3.73 | 1.17 | .05 | .06 | .84 | 3.55 | 1.19 | .16** | .14** |
| Self-other overlap (IOS) | - | 4.28 | 1.56 | .09 | -.08 | - | 3.47 | 1.91 | .19** | .09 |
| Conservatism | - | 3.18 | 1.49 | -.05 | -.21** | - | 4.03 | 1.89 | -.24** | -.24** |

COVID = COVID-19 vaccination intention, Dysomeria = hypothetical child vaccination intention, 5C = 5C Antecedents of Vaccination, IOS = inclusion of other (acquaintances for Study 1, community members for Study 2) in the self. Past flu vaccine uptake is a dummy variable with receiving the seasonal flu shot coded as 1. COVID and Dysomeria columns report correlations with other variables. Bolded $p < .05$

* $p < .01$

** $p < .001$.

COVID-19 and dysomeria vaccination intentions. Analytical thinking was positively associated with dysomeria but not COVID-19 vaccination intentions, and political conservatism was negatively associated with dysomeria but not COVID-19 vaccination intentions. Bivariate associations were non-significant for perceived disease vulnerability and self-other overlap in both types of vaccination intentions.

**Regression analyses.** All measures were standardized prior to analyses. To examine which constructs uniquely predicted vaccination intentions, in two separate models, we regressed COVID-19 and dysomeria vaccination intentions on all 14 measures. We used semi-partial $r$s (the unique contribution of each predictor; that is, the correlation between the outcome and the part of the focal predictor that is uncorrelated with the other predictors in the model) as estimates of predictor effect size. We then added gender, ethnicity, and log-transformed COVID U.S. deaths as control variables. Results are shown in Models 1a-1b (COVID-19) and Models 2a-2b (dysomeria) in Table 2.

The adjusted $R^2$ of these models were 21% (COVID-19) and 33%-34% (dysomeria), suggesting that the set of predictors, together, explained substantial variance in vaccination intentions. Past flu vaccination behavior (semipartial $r = .11$), vaccine confidence (semipartial $r = .16$), and collective responsibility (semipartial $r = .13$) emerged as unique predictors of COVID-19 vaccination intentions with and without control variables. Collective responsibility (semipartial $r = .22$), calculation (semipartial $r = -.10$), perceived vaccine danger (semipartial $r = -.17$), and analytic thinking (semipartial $r = .10$) uniquely predicted decisions regarding hypothetical child vaccine. Self-other overlap (semipartial $r = -.10$) was another unique

**Table 2. OLS regression models predicting vaccination intentions in Study 1 (undergraduates).**

| | COVID-19 Vaccine | | | Hypothetical Child Vaccine | | |
| | Model 1a | | Model 1b | Model 2a | | Model 2b |
| Predictor | β | sr | β | B | sr | β |
|---|---|---|---|---|---|---|
| Past vaccination behavior | **.25** | .11 | **.23** | .00 | .00 | -.01 |
| (1 = flu vaccine received) | [.03, .47] | | [.01, .46] | [-.20, .20] | | [-.21, .20] |
| Vaccine knowledge | .05 | .04 | .06 | -.01 | -.01 | -.01 |
| | [-.07, .17] | | [-.06, .18] | [-.12, .10] | | [-.12, .10] |
| 5C: Confidence | **.22**\* | .16 | **.23**\* | .09 | .07 | .10 |
| | [.09, .36] | | [.09, .37] | [-.03, .22] | | [-.03, .23] |
| 5C: Collective | **.18** | .13 | **.18** | **.30**\*\* | .22 | **.29**\*\* |
| | [.04, .31] | | [.04, .32] | [.17, .42] | | [.17, .42] |
| 5C: Complacency | .01 | .01 | .01 | -.11† | -.08 | -.10 |
| | [-.12, .14] | | [-.12, .15] | [-.23, .02] | | [-.22, .02] |
| 5C: Constraint | .02 | .02 | .02 | .00 | .00 | .00 |
| | [-.10, .14] | | [-.09, .14] | [-.11, .10] | | [-.11, .11] |
| 5C: Calculation | -.10† | -.09 | -.10† | **-.11** | -.10 | **-.11** |
| | [-.21, .01] | | [-.21, .02] | [-.21,-.01] | | [-.22, -.01] |
| Conspiracy belief | -.07 | -.05 | -.08 | .08 | .05 | .08 |
| | [-.22, .08] | | [-.23, .07] | [-.06, .22] | | [-.06, .22] |
| Vaccine danger | .02 | .01 | .01 | **-.28**\*\* | -.17 | **-.28**\*\* |
| | [-.14, .19] | | [-.15, .18] | [-.43, -.13] | | [-.43, -.13] |
| Science mistrust | -.05 | -.04 | -.05 | .10 | .07 | .10 |
| | [-.18, .08] | | [-.18, .08] | [-.02, .22] | | [-.02, .22] |
| Analytic thinking | -.03 | -.03 | -.02 | **.11** | .10 | **.11** |
| | [-.14, .08] | | [-.14, .09] | [.01, .21] | | [.01, .21] |
| Disease vulnerability | .03 | .03 | .03 | .01 | .01 | .01 |
| | [-.07, .14] | | [-.08, .14] | [-.09, .11] | | [-.09, .11] |
| Self-other overlap | .07 | .07 | .08 | **-.10** | -.10 | **-.10** |
| | [-.03, .17] | | [-.02, .19] | [-.19, -.01] | | [-.20, -.01] |
| Conservatism | .05 | .05 | .06 | -.04 | -.04 | -.04 |
| | [-.06, .16] | | [-.05, .18] | [-.14, .06] | | [-.14, .07] |
| Gender (1 = female) | | | .07 | | | .06 |
| | | | [-.19, .33] | | | [-.18, .30] |
| Ethnicity (1 = non-White) | | | .14 | | | -.02 |
| | | | [-.08, .37] | | | [-.23, .18] |
| COVID US deaths (log) | | | -.06 | | | |
| | | | [-.16, .04] | | | |
| N | 308 | | 307 | 308 | | 307 |
| R² /Adjusted R² | 0.25/0.21 | | 0.26/0.21 | 0.37/0.34 | | 0.37/0.33 |

Coefficients [95% CI] are standardized. *sr* = semipartial *r*.

† *p* < .10, bolded *p* < .05

\* *p* < .01

\*\* *p* < .001.

predictor for dysomeria, but the association was in the opposite direction than our prediction. Thus, although similar associations were observed for both types of vaccines in bivariate correlations, only collective responsibility emerged as a unique predictor for both vaccines in Study 1.

## Study 2

### Participants and procedure

Study 2 was a replication and extension of Study 1, remedying some technical issues from the first study and collecting data from a more age-diverse sample of online panel workers. Study 2 was preregistered at https://osf.io/rycjd. We conducted an a priori power analysis (see pre-registration) and aimed to collect complete data from a sample of 652 CloudResearch Prime Panel participants. Study 2 data were collected on July 20, 2020. We received 848 complete responses and after applying preregistered exclusion criteria (e.g., incomplete responses, attention check failures; see the S1 File for details), the final sample size was 676. Sensitivity analyses showed that with $\alpha = .05$ (two-tailed), we had 80% power to detect $r = |0.11|$. Procedures were identical to Study 1, except the 5C measures were assessed after the vaccination intention items in Study 2, but the order of remaining questionnaire was the same.

Participants were 62.87% female with mean age of 51.95 ($SD = 18.05$). Reported racial/ethnic identities were Asian/Asian American (6.07%), Black/African American (6.07%), Hispanic/Latino (4.14%), Native American/Pacific Islander (1.04%), White/European American (79.29%), more than one (1.78%), and other/prefer not to say (1.63%). Participants' annual household income was as follows: $0-$25,000 (21.30%), $25,001-$50,000 (26.48%), $50,001-$75,000 (19.97%), $75,001-$100,000 (14.35%), $100,001-$125,000 (6.21%), $125,001-$150,000 (4.88%), $150,001-$175,000 (1.92%), $175,001-$200,000 (1.78%), and more than $200,000 (3.11%). Additionally, 60.21% reported being a parent. When asked if they had been tested for COVID-19, the majority (88.61%) responded "No," 0.44% preferred not to respond, and 10.95% responded "Yes." Current residence of this sample included all 50 states (see S5 Table in S1 File). Demographic characteristics of Prime Panel samples tend to be more comparable to a nationally-representative sample than Amazon Mechanical Turk [29].

### Measures

The same measures from Study 1 were used in Study 2 with a few exceptions we note below.

**Vaccination intention (COVID-19).** Participants read CDC-provided information about COVID-19, including total U.S. cases and deaths and that there were no antiviral treatments or vaccines yet available (see S1 File for details). COVID-19 vaccination intentions were assessed with one discrete measure, followed by a continuous measure. Participants were first asked whether they would receive the COVID-19 vaccine if it were available, with response options: A = *I would not receive the vaccination even if it's free ($0)*, B = *I would receive the vaccination only if it's free ($0); if I need to pay money, I would not receive the vaccination*, and C = *I would pay money to receive the vaccination*. On the next page, participants moved a slider scale ($0-$500) to indicate the maximum amount of money they would personally pay for the vaccine.

As an exploratory measure, participants were also shown a list of eight common concerns (e.g., "The vaccine is too new") about the vaccine and were asked to indicate their agreement with each statement (1 = *Strongly disagree* to 7 = *Strongly agree*) and whether or not each concern mattered for their decision regarding COVID-19 vaccination [30].

**Past flu vaccination uptake.** Participants indicated their past flu vaccination uptake by responding to the question: "This past flu season (October 2019-April 2020), did you receive the flu shot?" Responses were coded such that 0 = No and 1 = Yes.

**Self-other overlap.** The same self-other overlap scale as Study 1 was used except the "other" circle represented other community members instead of acquaintances.

## Results

**Descriptive statistics.** Hesitancy toward COVID-19 vaccine was relatively high in Study 2 with 18.05% selecting A (I would not receive the vaccination even if it's free), 27.51% selecting B (if I need to pay money, I would not receive the vaccination), and only 54.44% selecting C (I would pay money to receive the vaccination). The maximum amount of money participants indicated they would personally pay for a COVID-19 vaccine varied substantially, ranging from $0 to $500 (M = 102.81, SD = 142.50, Median = 40.00).

We used the discrete measure and continuous measure to create a new variable similar to Study 1's measure, where 1 = selected option A and $0 on the slider, 2 = selected option B and $0 on the slider, 3 = $1-$10 on the slider, 4 = $11-$25, 5 = $26-$50, 6 = $51-$100, and 7 = indicated more than $100 on the slider. With this variable, 10.95% were willing to pay $0 and refused (i.e., they would not receive the vaccination even if it were free), 6.80% were willing to pay $0 and hesitant (i.e., they would not receive the vaccination if not free), and the amount of money other participants were willing to pay was: $1-$10 (11.09%), $11-$25 (17.01%), $26-$50 (16.57%), $51-$100 (15.09%), and more than $100 (22.49%). This combined measure of COVID-19 vaccination intention was used in correlational and regression analyses. Supplementary analyses treating the two items as separate measures are reported in the S1 File.

In Study 2, participants were asked about various concerns related to the COVID-19 vaccine. Table 3 shows the means and standard deviations of ratings for these concerns as a function of the discrete COVID-19 vaccination intention measure. When asked whether or not each concern mattered for participants' decision to vaccinate, "I worry about the side effects" (45%), "the vaccine is too new" (44%), and "the vaccine will not protect me" (35%) were the top three concerns (see S3 Fig in S1 File). These results corroborate other recent work on this topic [30].

For the hypothetical dysomeria vaccine, 6.66% indicated that they would "definitely not vaccinate" the child, and half of the sample (49.26%) endorsed the highest end of the scale that they would "definitely vaccinate" the child. For prior vaccination behavior, 55.77% reported receiving the seasonal flu shot.

**Bivariate correlations.** Table 1, presented earlier, shows Study 2's means, standard deviations, and correlations between the vaccination intention variables (COVID and dysomeria) and predictor variables. The full correlation matrices of all measured variables are available in the S1 File. The correlation between COVID-19 and dysomeria vaccination intentions was moderately high (r = .45, p < .001). Consequently, the correlational analyses revealed similar

**Table 3. Means and standard deviations of ratings for concerns influencing vaccination as a function of COVID-19 vaccination intention in Study 2.**

| COVID-19 Vaccination Intention | Refusal | | Hesitant | | Willing to pay | |
|---|---|---|---|---|---|---|
| Concern | M | SD | M | SD | M | SD |
| The vaccine is too new | 6.01 | 1.41 | 4.91 | 1.70 | 3.92 | 1.90 |
| I worry about the side effects | 5.89 | 1.68 | 4.95 | 1.80 | 3.90 | 1.96 |
| The vaccine will not protect me | 4.70 | 1.73 | 3.29 | 1.60 | 2.39 | 1.54 |
| I avoid most vaccines | 4.37 | 2.22 | 2.65 | 1.80 | 1.80 | 1.48 |
| COVID-19 is not severe enough to concern me | 3.93 | 2.15 | 2.46 | 1.80 | 1.68 | 1.39 |
| A doctor has recommended no vaccines | 3.58 | 2.12 | 3.45 | 1.97 | 2.86 | 2.11 |
| I will not have access to the vaccine | 3.02 | 1.78 | 3.41 | 1.65 | 2.66 | 1.58 |
| My religion prevents vaccination | 2.00 | 1.63 | 1.54 | 1.26 | 1.54 | 1.32 |

Refusal = "I would not receive the vaccination even if it's free ($0)." Hesitant = "I would receive the vaccination only if it's free ($0); If I need to pay money, I would not receive the vaccination." Willing to pay = "I would pay money to receive the vaccination."

results for both types of vaccination in Study 2. Notably, results from Study 2 mostly replicated those from Study 1 with a few exceptions (see Table 1).

As in Study 1, correlational analyses support most of our hypotheses. Specifically, past flu vaccine uptake, vaccine knowledge, vaccine confidence, collective responsibility, analytical thinking, disease vulnerability, and self-other overlap consistently emerged as positive correlates of COVID-19 and dysomeria vaccination intentions, although the effect sizes for the last three variables were somewhat modest (see Table 1). Complacency, anti-vaccine conspiracy beliefs, perceived vaccine danger, and mistrust in science/scientists were all negatively associated with COVID-19 and dysomeria vaccination intentions. Notably, political conservatism was negatively associated with COVID-19 vaccination in Study 2 but not in Study 1, which may be indicative of increased polarization of opinions regarding COVID-19 vaccination over time [19] or differences between student and online sampling frames. Constraint was negatively associated with dysomeria but not COVID-19 vaccination intentions. Calculation had non-significant associations with both types of vaccines in Study 2.

**Regression analyses.** All measures were standardized prior to analyses. Analytical methods were the same as Study 1, except we included gender, ethnicity, age, income, log-transformed COVID state deaths, and parenthood as additional demographic controls. Results are shown in Models 1a-1b (COVID-19) and Models 2a-2b (Dysomeria) in Table 4. The adjusted $R^2$ of these models were 30%-38% (COVID-19) and 39%-41% (dysomeria), suggesting that the set of predictors explained substantial variance in vaccination intentions.

Replicating Study 1, past flu vaccination behavior (semi-partial $r = .11$) and vaccine confidence (semipartial $r = .23$) emerged as unique predictors of COVID-19 vaccination intentions with and without control variables. Although collective responsibility was another unique predictor in Study 1, this trend was weaker in Study 2 with semi-partial $r = .04$. However, disease vulnerability (semipartial $r = .09$), self-other overlap (semipartial $r = .08$), and conservatism (semipartial $r = -.07$) emerged as unique predictors of COVID-19 vaccination intention in Study 2. Replicating Study 1, collective responsibility (semipartial $r = .09$) again emerged as a significant unique predictor of dysomeria vaccination intention, with and without controls. Additionally, vaccine knowledge (semipartial $r = .09$), confidence (semipartial $r = .24$), and complacency (semipartial $r = -.07$) each uniquely predicted dysomeria vaccination intention and were robust to inclusion of control variables. Vaccine confidence and collective responsibility—though latter to a lesser extent—were unique predictors of intentions for both vaccines in Study 2. Moreover, gender and income uniquely predicted intentions for both types of vaccines in Study 2 (see Table 4)—a finding consistent with other recent COVID-19 research [14, 17].

## Discussion

Although COVID-19 vaccines have become widely available in the U.S., vaccine hesitancy and refusal persist [31]. Hence, combatting vaccine hesitancy remains an extraordinary challenge for public health experts, and understanding the psychological roots of vaccine hesitancy during the ongoing pandemic is important to prepare for similar health crises in the future [32]. This work therefore remains timely in its identification of correlates of vaccine hesitancy, particularly since the rate of COVID-19 vaccination has begun to slow [33]. In the current research, a number of variables were associated with both actual (COVID-19) and hypothetical child (dysomeria) vaccine hesitancy.

### Key findings

Among both U.S. undergraduates (Study 1) and online panel workers (Study 2), we found that vaccination intentions were positively associated with past flu vaccination behavior, self-

**Table 4. OLS regression models predicting vaccination intentions in Study 2 (online panel workers).**

| Predictor | COVID-19 Vaccine | | | Hypothetical Child Vaccine | | |
| --- | --- | --- | --- | --- | --- | --- |
| | Model 1a | | Model 1b | Model 2a | | Model 2b |
| | β | sr | β | β | sr | β |
| Past vaccination behavior | **.25**\* | .11 | **.24**\*\* | .00 | .00 | .03 |
| (1 = flu vaccine received) | [.10, .40] | | [.10, .38] | [-.14, .14] | | [-.11, .17] |
| Vaccine knowledge | .04 | .03 | .00 | **.11**\* | .09 | **.09** |
| | [-.04, .11] | | [-.07, .07] | [.04, .19] | | [.02, .16] |
| 5C: Confidence | **.32**\*\* | .23 | **.27**\*\* | **.35**\*\* | .24 | **.32**\*\* |
| | [.23, .42] | | [.18, .35] | [.26, .43] | | [.23, .41] |
| 5C: Collective | .06 | .04 | .09† | **.13**\* | .09 | **.13**\* |
| | [-.03, .16] | | [.00, .19] | [.04, .23] | | [.04, .23] |
| 5C: Complacency | .05 | .03 | .03 | **-.12** | -.07 | **-.14**\* |
| | [-.06, .15] | | [-.07, .13] | [-.22, -.03] | | [-.23, -.04] |
| 5C: Constraint | **.10** | .08 | .06 | .03 | .02 | -.02 |
| | [.02, .18] | | [-.02, .14] | [-.05, .11] | | [-.10, .06] |
| 5C: Calculation | -.04 | -.04 | -.03 | .01 | .01 | .03 |
| | [-.11, .03] | | [-.10, .03] | [-.05, .07] | | [-.03, .10] |
| Conspiracy belief | .04 | .02 | .00 | -.09 | -.05 | **-.12** |
| | [-.08, .16] | | [-.11, .12] | [-.21, .02] | | [-.23, -.01] |
| Vaccine danger | -.10 | -.05 | -.04 | .00 | .00 | .02 |
| | [-.22, .02] | | [-.15, .07] | [-.11, .11] | | [-.09, .13] |
| Science mistrust | -.06 | -.04 | -.08 | -.03 | -.02 | -.02 |
| | [-.16, .04] | | [-.17, .02] | [-.13, .06] | | [-.12, .07] |
| Analytic thinking | .03 | .03 | .00 | -.04 | -.03 | -.06† |
| | [-.04, .11] | | [-.07, .07] | [-.11, .03] | | [-.13, .01] |
| Disease vulnerability | **.10**\* | .09 | **.12**\*\* | .03 | .03 | .03 |
| | [.03, .17] | | [.06, .19] | [-.03, .10] | | [-.03, .10] |
| Self-other overlap | **.09** | .08 | **.07** | -.03 | -.03 | -.04 |
| | [.02, .15] | | [.00, .13] | [-.09, .03] | | [-.10, .03] |
| Conservatism | **-.08** | -.07 | **-.10**\* | -.04 | -.03 | -.05 |
| | [-.15, -.01] | | [-.17, -.03] | [-.11, .03] | | [-.12, .02] |
| Gender (1 = female) | | | **-.21**\* | | | **-.19**\* |
| | | | [-.35, -.08] | | | [-.32, -.05] |
| Ethnicity (1 = non-White) | | | .00 | | | **-.16** |
| | | | [-.16, .16] | | | [-.32, .00] |
| Age | | | -.02 | | | **-.11**\* |
| | | | [-.10, .05] | | | [-.18, -.03] |
| Income | | | **.26**\*\* | | | **.08** |
| | | | [.19, .32] | | | [.02, .14] |
| COVID state deaths (log) | | | -.01 | | | |
| | | | [-.07, .06] | | | |
| Parenthood (1 = parent) | | | | | | .04 |
| | | | | | | [-.09, .17] |
| N | 674 | | 670 | 674 | | 670 |
| R² /Adjusted R² | 0.32/0.30 | | 0.40/0.38 | 0.40/0.39 | | 0.42/0.41 |

Coefficients [95% CI] are standardized. $sr$ = semipartial $r$.

† $p < .10$, bolded $p < .05$

\* $p < .01$

\*\* $p < .001$.

reported vaccine knowledge, confidence, and sense of collective responsibility. Complacency, anti-vaccine conspiracy beliefs, perceived vaccine danger, and mistrust in science/scientists were negative correlates. For constraint, calculation, analytic thinking, perceived disease vulnerability, self-other overlap, and political conservatism, strong associations were not consistently observed across vaccine types and samples. Consistent with prior research [30, 34, 35], previous vaccination against seasonal influenza was uniquely and strongly associated with COVID-19 vaccination intention. Confidence in the safety/effectiveness of vaccines also emerged as a unique predictor. Study 2 further revealed that worries about side effects, the vaccine being too new, and the vaccine not being protective were most prevalent concerns regarding the COVID-19 vaccine.

## Implications and contributions

To the extent that people who receive flu shots are more likely to also accept the COVID-19 vaccine, interventions promoting seasonal influenza vaccination in advance of a future health crisis may be one way to increase pandemic vaccine uptake. Given the strong association of vaccine confidence to intention, another useful avenue for intervention, particularly for a novel vaccine, may be to target confidence in the product by emphasizing its safety and highlighting a positive ratio of benefits to potential side effects, and communicating these points effectively using trusted parties. Although not all variables uniquely predicted vaccine hesitancy when accounting for shared variance among the larger set, individual predictors identified in zero-order correlations should not be discounted. Individually, collective responsibility, complacency, vaccine knowledge, conspiracy beliefs, perceived vaccine danger, and science mistrust were consistently associated with COVID-19 vaccination intentions in both studies. Given the strong overall predictive power of the variable set as a whole, it appears that there are multiple routes to tackling hesitancy.

To our knowledge, no prior study has simultaneously measured vaccine attitudes for both a hypothetical disease and a real disease during a pandemic. Measuring both together is important for providing context for past work regarding how well they generalize to actual intentions during a crisis period. That is, studying vaccination attitudes about unknown, hypothetical diseases [4, 5, 11] lacks ecological validity, and studying vaccination intentions for real, more "established" diseases for which vaccines have existed for years (and most people are inoculated) may not provide the same insights as attitudes towards a novel pandemic vaccine. The current work overcame these limitations and shows that factors associated with immunization decisions for a hypothetical disease are also mostly predictive of real vaccination intentions for a novel disease. By asking about both hypothetical and real diseases, this research has improved our understanding of the utility of previous work in its application to the current pandemic.

## Limitations

We acknowledge a few limitations of the current work. First, we cannot infer causality from our cross-sectional studies. Although both studies were conducted during critical times in the pandemic when effective treatments or vaccines were not yet available, longitudinal studies examining changes in vaccination attitudes across the pandemic timeline would be informative. Second, our samples of college students and online panel workers are not representative of all U.S. residents. Although a larger and nationally-representative sample is needed to capture COVID-19 vaccination intentions of the general U.S. population, we expect our results to be reproducible with different samples of students from similar undergraduate subject pools and Prime Panel workers. Moreover, understanding attitudes about pandemic vaccines and identifying potential reasons for hesitancy, even among a subset of the populations, is critical

to address the broader goal of achieving sufficient vaccine coverage. Third, COVID-19 vaccine intention was operationalized as self-reported willingness to pay money for the vaccine. Although this item should reflect commitment to vaccinate, future studies could benefit from measuring actual delay in vaccinating or vaccination behavior as indicators of hesitancy. Despite these limitations, the present work documents psychological factors associated with vaccine hesitancy in the context of a life-changing pandemic.

## Supporting information

**S1 File. Online supplementary materials.** Study materials, supplementary measures and analyses.
(DOCX)

## Author Contributions

**Conceptualization:** Suryaa Gupta, Shoko Watanabe.

**Data curation:** Shoko Watanabe.

**Formal analysis:** Shoko Watanabe, Sean M. Laurent.

**Funding acquisition:** Suryaa Gupta, Shoko Watanabe, Sean M. Laurent.

**Investigation:** Suryaa Gupta, Shoko Watanabe.

**Methodology:** Suryaa Gupta, Shoko Watanabe, Sean M. Laurent.

**Project administration:** Shoko Watanabe, Sean M. Laurent.

**Resources:** Sean M. Laurent.

**Supervision:** Sean M. Laurent.

**Visualization:** Shoko Watanabe.

**Writing – original draft:** Suryaa Gupta.

**Writing – review & editing:** Shoko Watanabe, Sean M. Laurent.

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
