## [Decision Letter · Decision Letter 0]

1 Oct 2021

PONE-D-21-22539Psychological predictors of vaccination intentions among U.S. undergraduates and online panel workers during the 2020 COVID-19 PandemicPLOS ONE

Dear Dr. Watanabe,

Thank you for submitting your manuscript to PLOS ONE. After careful consideration, we feel that it has merit but does not fully meet PLOS ONE’s publication criteria as it currently stands. Therefore, we invite you to submit a revised version of the manuscript that addresses the points raised during the review process.

We look forward to receiving your revised manuscript.

Kind regards,

Ismaeel Yunusa, PharmD, PhD

Academic Editor

PLOS ONE

Journal Requirements:

Reviewers' comments:

Reviewer's Responses to Questions

**Comments to the Author**

1. Is the manuscript technically sound, and do the data support the conclusions?

Reviewer #1: Partly

Reviewer #2: Yes

2. Has the statistical analysis been performed appropriately and rigorously? 

Reviewer #1: Yes

Reviewer #2: No

3. Have the authors made all data underlying the findings in their manuscript fully available?

Reviewer #1: Yes

Reviewer #2: Yes

4. Is the manuscript presented in an intelligible fashion and written in standard English?

Reviewer #1: Yes

Reviewer #2: No

5. Review Comments to the Author

Reviewer #1: This manuscript adds an increasing list of similar researches on psychological predictors of covid-19 vaccination intentions, and the results are not surprisingly consistent with what others have been found, but still it's good to see the authors collected two new samples on vaccine intentions (the college undergrads and the online workers).

My main questions are:

1) There is no justification of the representativeness of the sample for their surveys relative to the general US populations, which makes it hard to assess whether the study is reliably capturing the attitudes of the general population toward covid-19 vaccination. The sample size for each of the two samples seemed too small to be much representative.

2) There is a concern that some of the reported statistical analyses may not be valid, due to likely violations of the assumptions of a linear regression model. This is especially for the college student sample. If the students surveyed have certain clustering effect, e.g., some are in the class, in the same department or program, then the data is not independent. If this is the case, some kind of linear mixed model with random effects is more appropriate than the multiple regression model that the authors employed.

Reviewer #2: Thank you for the opportunity to review this manuscript. I appreciate the amount of work that has been put into this study; the authors should be commended for their efforts to study the psychological roots of vaccine hesitancy during the current pandemic. Below are my concerns

•I have major problems with the way the data are presented, and the manuscript is written and I recommend major revision to have a more concise manuscript with a clear flow and more clear conclusion. I include a couple of examples on issues that need to be further clarified below.

•I do not see any benefit of splitting the study samples into 1& 2, and if there was, it was not made clear in the presentation of the study. I understand that sample 2 was needed to have a more age-diverse sample, but why not combine them? If it was because you expected differences between the two samples, then explain why you would put this assumption? On what basis? This needs to be explained or changed. There are also differences in the questions asked to each sample which were not explained; on what basis did the author choose to do so? Any benefits? Wouldn't that increase variability of the results among the samples?

•The manuscript reads more like a thesis than a concise manuscript it lacks “straight to the point” presentation of data and clear conclusions. I recommend rewriting the manuscript, especially the introduction, which should be one coherent piece of work, also the methods and the results. The conclusion in the abstract will need reconsideration as it starts with a vague sentence and does not give any concise conclusion

•The questionnaires used are very lengthy, and despite the attention checks the authors have described, no information has been provided regarding how many participants were excluded due to failing the attention check; they only report a general total number of 38 excluded for different reasons for sample one and same regarding sample 2 were 172 were excluded for several reasons.

•Line 85: "Whereas previous research has examined attitudes about vaccination 86 for diseases that are hypothetical (e.g., Haase et al., 2015; Jolley & Douglas, 2017), relatively 87 low-threat (e.g., seasonal flu; Zhang et al., 2010) or mostly eradicated in the U.S. (e.g., Dubé et 88 al., 2015), the present research investigates vaccination intentions during a unique period of uncertainty involving a highly contagious disease when effective treatments or vaccines had not yet been developed." The authors should report the published evidence on vaccine intentions in the COVID-19 era

•The authors mention that” Sample 1 included 346 undergraduate students from the University of Illinois at Urbana183 Champaign recruited in exchange for course credit” Can you elaborate more on what do you mean here? Were the participants given a choice to join? How did you guarantee that participants were aware that their participation is optional?

•The discussion section can be improved by highlighting the take-home message from the study and take the reader through what has been done and what does that mean rather than displaying the results. As an example in line 418: "knowledge, no prior study has simultaneously measured vaccine attitudes for both a hypothetical disease and a real disease during a pandemic. Measuring both together and determining which variables predict well across measures is important, as it provides context for past work and can bolster confidence in how well past research applies to actual intentions during a crisis period…… The current work shows that factors associated with immunization decisions for a hypothetical disease are also mostly predictive of real vaccination intentions." I do not see the point here? Can you elaborate on the reasons behind studying vaccine attitudes for both hypothetical disease and a real disease during a pandemic? Not at any point of the manuscript, this was discussed clearly

•The tables are confusing with too many details; I suggest editing them to highlight the most important message from each table and move any unnecessary information to the supplementary file.

•In the supplementary file, Why was the attention check different between samples 1 &2? What did you base this on?

For Sample 1, the following item was asked prior to the demographics questionnaire:

•Thank you for your responses. You are almost finished with this study. Before answering a few questions about yourself, we have one last question. Do you think we should include your responses in our study? That is, did you take the study seriously and respond thoughtfully? Your credit assignment for this study does not depend on your response to this question. [Yes, I responded thoughtfully / No, I did not respond thoughtfully.]

Sample 2 Only

For Sample 2, the following item was embedded in the optimism bias measure:

If you are reading this, please select 67 on

this sliding scale.

6. PLOS authors have the option to publish the peer review history of their article (what does this mean?). If published, this will include your full peer review and any attached files.

Reviewer #1: No

Reviewer #2: No

---

## [Author Response · Author response to Decision Letter 0]

7 Nov 2021

Author Response to Reviewer Comments

Thank you for offering us the opportunity to revise and resubmit our manuscript “Psychological predictors of vaccination intentions among U.S. undergraduates and online panel workers during the 2020 COVID-19 Pandemic” to PLOS ONE. We sincerely thank you for the thoughtfulness and care with which you read and evaluated our work. Our detailed responses to your comments can be found below, in bold blue print. Thank you again for your time.

Comments from Reviewer 1

1. This manuscript adds an increasing list of similar researches on psychological predictors of covid-19 vaccination intentions, and the results are not surprisingly consistent with what others have been found, but still it's good to see the authors collected two new samples on vaccine intentions (the college undergrads and the online workers).

RESPONSE: Thank you for your positive evaluation of our manuscript. We agree that COVID-19 vaccine intention is an interesting, important, and growing area of research.

2. My main questions are:

1) There is no justification of the representativeness of the sample for their surveys relative to the general US populations, which makes it hard to assess whether the study is reliably capturing the attitudes of the general population toward covid-19 vaccination. The sample size for each of the two samples seemed too small to be much representative.

Response: Thank you for bringing up this important point regarding the representativeness of our samples. We agree that our samples of undergraduate students and online panel workers are not necessarily representative of the U.S. population. In the original submission, we had acknowledged this limitation in the General Discussion on p. 23 (line 434-435): “…the present study was conducted online in the U.S. using convenience samples, which were not representative of all U.S. residents.” Based on your feedback, we have discussed this limitation further on p. 25 (lines 495-501) of the revised manuscript and made this more salient by creating a “limitations” section. While we continue to acknowledge that our samples are not necessarily representative of the general U.S. population, we have added that we expect the results to be reproducible with students from similar undergraduate subject pools and with Prime Panel workers because our samples are representative of target populations from which we sampled. Moreover, we believe that identifying potential reasons for hesitancy, even among a subset of the population, is critical to address the broader goal of achieving sufficient vaccine coverage. Additionally, we noted on pp. 17 (lines 335-337) that our online sample was drawn from all 50 states and that Prime Panels are more representative of the country than other online platforms (e.g., Amazon Mechanical Turk) across demographic variables including age, marital status, number of children, political affiliation, and religious devotion (Chandler et al., 2019). 

3. 2) There is a concern that some of the reported statistical analyses may not be valid, due to likely violations of the assumptions of a linear regression model. This is especially for the college student sample. If the students surveyed have certain clustering effect, e.g., some are in the class, in the same department or program, then the data is not independent. If this is the case, some kind of linear mixed model with random effects is more appropriate than the multiple regression model that the authors employed.

Response: In Study 1, all participants were recruited from an undergraduate psychology course-credit subject pool at the University of Illinois Urbana-Champaign in Spring 2020. Undergraduate students who are 18 or older taking one or more psychology courses have the option to enroll in the subject pool to participate in research studies for course credit or to complete an alternative assignment. Participants in the subject pool individually sign up to complete studies on their own time (i.e., outside of class). Although participants in the subject pool consist of students with different majors, year in college, classes, etc., a) our data represent individual responses of these students who are all drawn from same participant pool, and b) we did not collect data on students’ home department, year in college, classes taken, etc. Hence, no clustering effect can be modeled. As we discuss in our response to Reviewer 2’s comment regarding the subject pool (#10 below), we additionally clarify in the revised manuscript on p. 8 (lines 175-181) that data collection using undergraduate subject pools is a standard practice in psychology and social sciences more broadly (Rossell et al., 2005; Sadeghiyeh et al., 2021). 

Comments from Reviewer 2

4. Thank you for the opportunity to review this manuscript. I appreciate the amount of work that has been put into this study; the authors should be commended for their efforts to study the psychological roots of vaccine hesitancy during the current pandemic. 

RESPONSE: Thank you. We appreciate your positive feedback as well as your critical insights that have allowed us to improve the work.

5. Below are my concerns:

I have major problems with the way the data are presented, and the manuscript is written and I recommend major revision to have a more concise manuscript with a clear flow and more clear conclusion. I include a couple of examples on issues that need to be further clarified below. 

RESPONSE: Thank you for this feedback and your detailed comments. In our revision, we believe we have addressed these concerns (see below). 

6. I do not see any benefit of splitting the study samples into 1& 2, and if there was, it was not made clear in the presentation of the study. I understand that sample 2 was needed to have a more age-diverse sample, but why not combine them? If it was because you expected differences between the two samples, then explain why you would put this assumption? On what basis? This needs to be explained or changed. There are also differences in the questions asked to each sample which were not explained; on what basis did the author choose to do so? Any benefits? Wouldn't that increase variability of the results among the samples?

Response: Thank you for your comments regarding the presentation of the data. In the original manuscript, we described Sample 1 as the undergraduate student sample and Sample 2 as the online panel worker sample rather than describe them as participants from two distinct studies because we sought to simplify the discussion of methods/measures (which were nearly identical) and results. Based on your comment, however, we have decided to present the data as two separate “studies” (Study 1 corresponding to the undergraduate sample and Study 2 corresponding to the Prime Panel sample). Although we did not hypothesize specific differences to emerge between the two samples, there are several notable differences between the two sets of data that warrant separate analyses. 

First, Study 2 was a pre-registered study intended to replicate the effects observed in Study 1 with a new sample. Second and relatedly, we fixed minor technical issues from the first survey and added new/modified items to Study 2. For example, our primary dependent measure of COVID-19 vaccination intention was asked with two items in Study 2 instead of using a single-item measure (see p. 17 lines 341-350). Third and most importantly, these studies were conducted at different time points during the Pandemic. Study 1 was conducted during the initial months of the Pandemic (February-April 2020) when there was much uncertainty and general lack of information about the virus (e.g., infectiousness, symptoms, etc.) and while U.S. cases were growing exponentially and lockdowns had just begun; in contrast, Study 2 was collected on July 20, 2020—roughly 6 months after the first U.S. laboratory-confirmed COVID case and cases were steadily growing. Notably, Study 2 was conducted during Phase II clinical trials toward the end of the first U.S. wave when more information about the virus and about vaccine development were available, and when people were likely starting to get used to the “new normal.” Because willingness to receive a COVID vaccine has fluctuated over time in the U.S. population with reported percentages varying substantially depending on when the poll was taken (e.g., Fridman et al., 2021; Funk & Tyson, 2020; Tyson et al., 2020), an analysis combining the two datasets may produce potentially misleading conclusions. 

In sum, for succinctness, we presented the results using measures that were collected for both samples, which may have given the impression that we “split” the sample. However, these data were collected in two distinct studies conducted at different points during the pandemic. To clarify that the data are distinct in time and sample characteristics, we now treat them as two separate “studies” in our revised manuscript and further emphasize that Study 2 is a replication of Study 1 (see p. 3 line 62, p. 16 lines 315-318). Moreover, we highlight the improvements made in the second survey based on the results from the first survey (see p. 10, lines 228-229) and include a new variable about concerns regarding COVID-19 vaccine (see p. 17 lines 351-354), which was originally described in the Online Supplementary Materials. Finally, we have restructured the manuscript such that participants, procedures, measures, and results for Study 1 and Study 2 are described separately. 

7. The manuscript reads more like a thesis than a concise manuscript it lacks “straight to the point” presentation of data and clear conclusions. I recommend rewriting the manuscript, especially the introduction, which should be one coherent piece of work, also the methods and the results. The conclusion in the abstract will need reconsideration as it starts with a vague sentence and does not give any concise conclusion

Response: Thank you for your feedback. We have made substantial revision to the manuscript following your suggestions. Specifically, the introduction now succinctly describes the objectives and motivation for the research. We deleted descriptions of related works that were not essential to the background of the current research, reducing the length of the introduction from 922 words to 606. As mentioned earlier, we have entirely restructured the methods and results section such that the two studies are now described separately. Moreover, we have added the “key findings” section in the General Discussion to highlight the take-home message (see p. 23 lines 451-463). We also revised the conclusion in the abstract, and now it read as follows: “Encouraging flu vaccine uptake, enhancing confidence in a novel vaccine, and fostering a sense of collective responsibility are particularly important as they uniquely predict COVID-19 vaccination intentions.” (pp. 3-4 lines 75-77).

8. The questionnaires used are very lengthy, and despite the attention checks the authors have described, no information has been provided regarding how many participants were excluded due to failing the attention check; they only report a general total number of 38 excluded for different reasons for sample one and same regarding sample 2 were 172 were excluded for several reasons. 

Response: Thank you for your feedback. As described in the supplementary file (OSM p. 10), there were two attention checks in Study 1. The first item asked: If you’re reading this, please check “Strongly Agree.” Out of 346 respondents, n=34 failed this question and selected a response other than “Strongly Agree.” The second item was displayed at the end of the survey and asked whether they took the study seriously and responded thoughtfully. Out of 346 respondents, n=10 responded that they did not take the study seriously. In summary, 28 respondents failed the first item only, 4 respondents failed the second item only, and 6 respondents failed both items, leaving us with the final sample size of N=346-28-4-6=308 consisting of participants who passed both attention check items. 

For Study 2, we had three pre-registered exclusion criteria: respondents who 1) do not finish the survey, 2) fail to answer one or more attention checks correctly, and 3) fail to spend more than 10 seconds on pages involving reading and/or responding to multiple items. We received 848 complete responses in Study 2. The first attention check item was identical to Study 1. Out of 848 respondents, n=117 failed to select “Strongly Agree.” The second attention check item asked respondents to select “67” on a sliding scale. Out of 848 respondents, n=109 provided a response other than 67. In summary, 58 respondents failed the first item only, 50 respondents failed the second item only, and 59 respondents failed both items, leaving us with n=848-58-50-59=681 respondents who passed both attention check items in Study 2. Finally, we checked the average reading time and excluded n=5 respondents who spent less than 10s per page. Thus, the final sample size for Study 2 is N=681-5=676. 

Because footnotes are not allowed in PLOS ONE and to further increase concision, we direct readers in the main text (p. 8 line 187, p. 16 lines 321-322) to see the supplementary file, which now contains the above details regarding exclusions for Study 1 (OSM p. 18) and Study 2 (OSM pp. 20-21). 

9. Line 85: "Whereas previous research has examined attitudes about vaccination for diseases that are hypothetical (e.g., Haase et al., 2015; Jolley & Douglas, 2017), relatively low-threat (e.g., seasonal flu; Zhang et al., 2010) or mostly eradicated in the U.S. (e.g., Dubé et 88 al., 2015), the present research investigates vaccination intentions during a unique period of uncertainty involving a highly contagious disease when effective treatments or vaccines had not yet been developed." The authors should report the published evidence on vaccine intentions in the COVID-19 era 

Response: Thank you for your suggestion. In this statement, we wanted to make a point that our research (as well as other research conducted during the pandemic) is different from past research on vaccine hesitancy conducted prior to the pandemic. We have modified this sentence as follows (p. 5, lines 111-116): “Whereas vaccine hesitancy research prior to the pandemic examined attitudes about vaccination for diseases that are hypothetical, relatively low-threat (e.g., seasonal flu), or mostly eradicated in the U.S. (e.g., polio), the present research examined which psychological factors are associated with intention to receive a novel, pandemic vaccine in a context where no one was immune…”

However, we agree that more research on vaccine hesitancy specific to COVID-19 has been published since we originally drafted this manuscript. Therefore, we have incorporated these recent works in the introduction (see pp. 6-7 lines 131-149), results (see p. 19 lines 387-388, p. 20 lines 412-415, p. 21 lines 438-439), and discussion (p. 23 lines 458-460). 

10. The authors mention that” Sample 1 included 346 undergraduate students from the University of Illinois at Urbana Champaign recruited in exchange for course credit” Can you elaborate more on what do you mean here? Were the participants given a choice to join? How did you guarantee that participants were aware that their participation is optional?

Response: In Study 1, all participants were recruited from an undergraduate psychology course-credit subject pool at the University of Illinois Urbana-Champaign in Spring 2020. Undergraduate students who are 18 or older taking one or more psychology courses have the option to enroll in the subject pool to participate in research studies for course credit or to complete an alternative assignment with similar duration. This information is provided in the course syllabus and the informed consent that participants read prior to starting the study. Participants in the subject pool individually sign up to complete studies on their own time. Participation is voluntary, and students are allowed to withdraw from the study for any reason without any penalty. We have clarified our recruitment procedure for Study 1 on p. 8 (lines 175-185) of the revised manuscript. Moreover, we note that data collection using undergraduate subject pools is a standard practice in psychology and related fields within the social sciences more broadly (e.g., business, consumer research, economics). 

11. The discussion section can be improved by highlighting the take-home message from the study and take the reader through what has been done and what does that mean rather than displaying the results. As an example in line 418: "knowledge, no prior study has simultaneously measured vaccine attitudes for both a hypothetical disease and a real disease during a pandemic. Measuring both together and determining which variables predict well across measures is important, as it provides context for past work and can bolster confidence in how well past research applies to actual intentions during a crisis period…… The current work shows that factors associated with immunization decisions for a hypothetical disease are also mostly predictive of real vaccination intentions." I do not see the point here? Can you elaborate on the reasons behind studying vaccine attitudes for both hypothetical disease and a real disease during a pandemic? Not at any point of the manuscript, this was discussed clearly

Response: Thank you for this suggestion. We have added a “key findings” section in the General Discussion to emphasize the take-home message on p. 23 (lines 451-463). In addition, we have revised the paragraph you referred to above (p. 24, lines 478-489) to clarify our point regarding the importance of measuring vaccination attitudes for both a hypothetical disease and a real disease. We argue that measuring both together is important for providing context for past work regarding how well they generalize to actual intentions during a crisis period. That is, studying vaccination attitudes about a novel yet hypothetical disease (e.g., Haase et al., 2015; Jolley & Douglas, 2017) lacks ecological validity, and studying vaccination intentions for real, more “established” diseases for which vaccines have existed for years (and most people are inoculated) may not provide the same insights as attitudes towards a novel pandemic vaccine. The current work overcame these limitations, also finding that factors associated with immunization decisions for a hypothetical disease are mostly predictive of real vaccination intentions. By asking about both hypothetical and real diseases, this research has improved our understanding of the utility of previous work in its application to the current pandemic. We also articulate this point earlier on in the manuscript (see pp. 5-6 lines 117-130). 

12. The tables are confusing with too many details; I suggest editing them to highlight the most important message from each table and move any unnecessary information to the supplementary file.

Response: Thank you for your feedback. We have moved the Table 1-2 from the original version to the supplementary file, and we now present a simpler version as Table 1 (p. 13). We bolded the significant correlations so that they are easily noticeable. Similarly, we have simplified the tables describing the regression models (Table 2 on p. 15 and Table 4 on p. 22). The original tables each contained results of 30 distinct models, but the revised tables each contain 4 only models. 

13. In the supplementary file, Why was the attention check different between samples 1 &2? What did you base this on?

For Sample 1, the following item was asked prior to the demographics questionnaire: Thank you for your responses. You are almost finished with this study. Before answering a few questions about yourself, we have one last question. Do you think we should include your responses in our study? That is, did you take the study seriously and respond thoughtfully? Your credit assignment for this study does not depend on your response to this question. [Yes, I responded thoughtfully / No, I did not respond thoughtfully.]

Sample 2 Only

For Sample 2, the following item was embedded in the optimism bias measure:

If you are reading this, please select 67 on this sliding scale. 

Response: We appreciate your inquiry regarding the attention checks and for taking the time to review the supplementary materials. There are three reasons why we decided to alter our approach in Study 2. First, we noticed from our previous work that has used online panel workers that almost all participants claim that they responded thoughtfully even when we explicitly tell them that their payment will not be affected on the basis of their response. Thus, we decided to drop this particular question which did not seem to be providing much information about actual attentiveness. Second, for a question with two response options (“Yes, I responded thoughtfully” vs. “No, I did not respond thoughtfully”), the chance of answering “Yes” is 50% even if a participant were to select a random response. With a slider scale like above, there is only 1% chance that a non-attentive respondent will accidentally move the slider to the correct number. To ensure obtaining quality data, we decided to have a stricter attention check item in Study 2. Third, there were other measures (including the main COVID-19 vaccination intention item) that used a slider scale in Study 2, whereas all questions were in multiple-choice or Likert-type scale formats in Study 1. An ancillary reason for using the slider scale as an attention check was to confirm that participants were able to indicate their desired response using the slider format. We have noted these reasons in the supplementary file (OSM p. 10). Finally, and importantly, we also note that these attention check items were specified in our pre-registration.

---

## [Editor Report · Decision Letter 1]

9 Nov 2021

Psychological predictors of vaccination intentions among U.S. undergraduates and online panel workers during the 2020 COVID-19 Pandemic

PONE-D-21-22539R1

Dear Dr. Watanabe,

We’re pleased to inform you that your manuscript has been judged scientifically suitable for publication and will be formally accepted for publication once it meets all outstanding technical requirements.

Kind regards,

Ismaeel Yunusa, PharmD, PhD

Academic Editor

PLOS ONE

Additional Editor Comments (optional):

Thank you for your contributions!
---

## [Editor Report · Acceptance letter]

17 Nov 2021

PONE-D-21-22539R1 

Psychological predictors of vaccination intentions among U.S. undergraduates and
online panel workers during the 2020 COVID-19 Pandemic 

Dear Dr. Watanabe:

I'm pleased to inform you that your manuscript has been deemed suitable for publication in PLOS ONE. Congratulations! Your manuscript is now with our production department. 

Kind regards, 

on behalf of

Dr. Ismaeel Yunusa 

Academic Editor

PLOS ONE